# Correlation Analysis of Real-Time Warning Factors for Construction Heavy Trucks Based on Electrified Supervision System

Weiwei Qi [1] , Shufang Zhu [1] and Jinsong Hu [2,*]

1   School of Civil Engineering and Transportation, South China University of Technology,
    Guangzhou 510641, China
2   Guangzhou Transport Planning Research Institute Co., Ltd., Guangzhou 510030, China
*   Correspondence: gztpri@126.com

**Abstract:** Due to inertia, heavy trucks are often involved in serious losses in accidents. To prevent such accidents, since 2020, the transportation department has promoted the free installation of intelligent video surveillance systems on key vehicles of "two passengers, one danger, and one cargo". The system can provide real-time warnings to drivers for various risky driving behaviors. The data collected by the system are often managed by third-party platforms, and such platforms do not have authority beyond the information that the authority system can collect. Therefore, it is necessary to use the trajectory data and warning behavior records that the system can collect for behavior analysis and accident prevention. To analyze the correlation between different warning factors, 88,841 warning records and 1033 trip records of heavy trucks for construction in the second half of 2021 were collected from a third-party supervision platform. The research associated the warning records with the vehicle operation records according to the warning time and the license plate and established a multiple linear regression equation associated with operational attributes and warning factors. The factor selection results showed that only two warning factors, "too close distance" and "lane change across solid line", can be used as dependent variables to construct a regression model. The results showed that many distracted behaviors had a significant impact on aggressive driving behavior. Companies need to focus on behaviors that are prone to other warning behaviors. This paper provides a theoretical basis for the optimization of the warning function of the electrified supervision system and the continuing education of drivers by exploring the internal correlation between different warning factors.

**Keywords:** heavy truck; electrified supervision system; multiple linear regression; lasso regression; optimal subset

## 1. Introduction

In 2020, electric buses were fully popularized in Guangzhou. With the popularization of electrified vehicles, electronic in-vehicle devices with real-time monitoring functions are also gradually popularizing in road transportation vehicles. With the continuous expansion of social construction needs, the number of trucks and their transportation mileage continue to increase. According to statistics [1], 11,128,000 trucks were owned as of 2020. Although road traffic continues to develop, the safety situation is not optimistic. According to statistics [1], 244,700 traffic accidents occurred in 2020. To eliminate potential accidents fundamentally, the management department attaches great importance to the supervision of the whole process of traffic operation during the construction of intelligent transportation system. "Two passengers, one danger, and one cargo" refers to chartered vehicles for tourism; three or more class of bus to work; special road vehicles and heavy trucks for transporting hazardous chemicals, fireworks and firecrackers; and civilian explosives. "Two

passengers, one danger and one cargo" vehicle accidents are often accompanied by serious losses and are an important supervision object for road transportation production.

To formulate scientific safety improvement programs, exploring the causes of truck traffic accidents is an important issue in the field of traffic safety research. There have been a lot of research results in the analysis of the causes of truck driver accidents. The data sources of related research are mainly divided into three categories: One is to conduct driving experiments based on simulated drivers [2] or simulation programs [3] and collect physiological indicators of drivers under hypothetical conditions and corresponding driving behavior. For example, through the collision procedure, experiments with different truck weights and bumper heights were carried out at different speeds to estimate the throw distance after collision with pedestrians [4]. The second is to analyze the influence of road, environment, traffic flow, and other factors within the research scope by building a statistical model based on historical collision data. Risky driving behaviors, such as aggressive driving, failure to keep to the proper lane [5], speeding, drowsiness and fatigue, distraction [6], and inattention have been shown to significantly affect the likelihood of a crash. As expected, seat belt use significantly decreases the severity of a run-off-road crash [7]. The environment, such as severe weather [8], wind speed, rainfall, humidity, air temperature [9], midnight [10], and other factors, significantly affects the accident rate. Influencing factors also include the driver's own characteristics, such as age [11], gender [12], and weight [13]. The third is to design a questionnaire that collects the behavioral intentions and personal characteristics of a certain group of truck drivers. For example, with the mediation of attitude toward risky driving, risk perception had a negative influence on the intention, and attitude toward risky driving had a positive influence on the intention [14]. Factors such as angry driving [15], compensation [16], and personality [14] have all been shown to influence truck drivers' risky driving behavior.

However, the cost of data collection for simulation experiments is high, and detailed data on historical accidents are generally not available to the public. The lack of third-party management companies with the above data makes it difficult to optimize risky driving-monitoring systems to strengthen the dynamic supervision of road transport vehicles, and prevent and reduce road transport accidents. Guangdong Province has formulated management measures for vehicles with "two passengers, one danger, and one cargo". First, road transportation enterprises need to install and use intelligent video monitoring and alarm devices that meet the technical standards of Guangdong Province for their passenger vehicles, dangerous goods vehicles, and heavy vehicles and report the monitoring data to the intelligent supervision system in real time through the main link of the equipment. The second is to encourage third-party monitoring agencies to provide professional monitoring services for road transport enterprises. A series of intelligent supervision systems is constructed to reduce the operational risk of road transport vehicles. The intelligent video monitoring and alarm device has functions such as satellite positioning, vehicle video monitoring, advanced driving assistance, and driver status monitoring.

When the vehicle is running, the system can monitor driving risks and bad driving behaviors in real time and issue warnings to the driver. Status warning identification is mainly through preset scenarios, and each warning factor is a separate identification module. When the warning is triggered, the warning event itself has occurred, such as changing lanes across a solid line. The electrified supervision system is part of an advanced driver assistance system (ADAS). Based on the results of one-way analysis of variance, most warnings from ADAS positively affected commercial truck drivers' behaviors [17], for example, on safety headway improvement, compliance with lane departure, keeping to the speed limit, and collision avoidance under high-speed driving conditions. The intervention of in-vehicle monitoring system (IVMS) has also been proven to be beneficial to reducing risky driving behavior [18]. The results showed that the combination of risk training can optimize the intervention effect of driving warning on risky driving. However, different alarming behaviors are always classified as risky behaviors, and the internal correlation between them lacks research.

To further optimize the warning system, it is necessary to explore more factors that lead to the occurrence of warnings, rather than relying only on identifying whether the event has occurred. The third-party supervision platform needs to regularly provide feedback on the operation report to the operating company based on the historical warning records. The purpose of this paper is to analyze the internal correlation between different warning behaviors based on the existing large number of alarm records and trip records of third-party-platform companies. The model was constructed using multiple linear regression equations that can intuitively explain the influence between variables. The optimal subset method was used to discover potential combinations of variables that could be used to build a model, and Lasso regression was used to reduce the number of model variables. Figure 1 is the framework of the research.

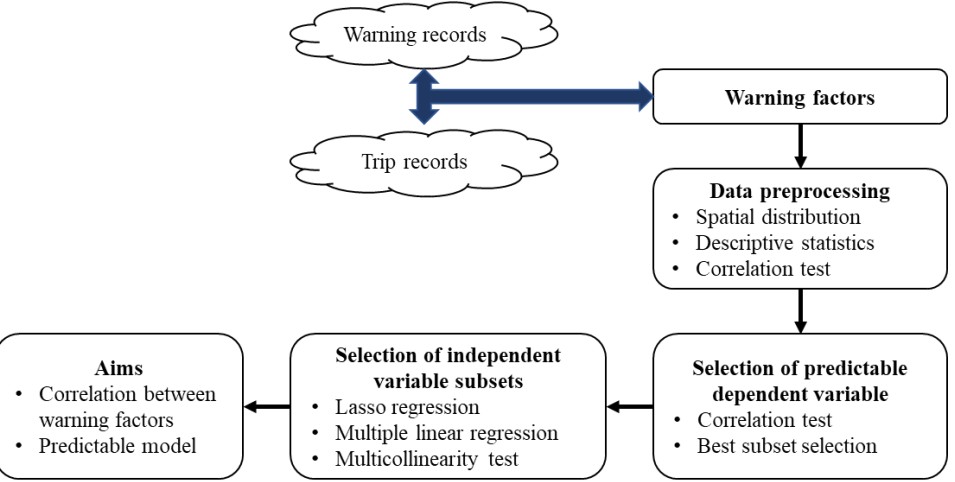

**Figure 1.** The framework of the research.

## 2. Literature Review

Heavy construction trucks generally refers to heavy trucks that transport soil and stone and other infrastructure materials for construction sites. Wages for such drivers are calculated by the number of trips and weight. Excessive driving and speeding are rampant for better pay. Due to the serious traffic accident losses of heavy construction trucks, the insurance of the vehicles is much higher than that of ordinary vehicles [19]. Research on heavy construction trucks is mainly divided into two categories. One is to study the optimization of scheduling to improve engineering efficiency [20]. Among them, the "Truck+" platform integrates a global positioning system (GPS) and geographic information system (GIS) through intelligent vehicle-mounted equipment. Implemented monitoring and forecasting productivity function for project teams [21]. Shehadeh et al. built an earthwork model based on a genetic algorithm to reduce the time and cost of earthwork [22]. The second is to study the impact of the driving process of heavy construction trucks on the health of drivers [23]. Burgess-Limerick and Lynas [24] confirmed that operators of dozers and off-road haul trucks in use at surface coal mines are frequently exposed to vertical whole-body vibration levels that lie within or above the Health Guidance Caution Zone defined by ISO2631.1.

With the popularity of driver-assistance systems, research has expanded to geospatial analysis based on geographic information data. A comparative analysis found that road sections greater than 20 miles from a rest area were more prone to accidents [25]. Compared with natural driving, the data obtained based on collision reconstruction has inherent limitations [26], and historical collision data are prone to missing records of key information. By loading the natural driving data collected by intelligent vehicle equipment, not only is the information collected when the event occurs, but the most realistic driving state can also be restored. Research in natural driving has shown that the warning of rear-end

collisions by in-vehicle collision warning systems enables drivers to react faster in the face of conflicts than without warning [27].

Due to the randomness and variability of collision accidents, scholars search for more frequent and more observable traffic characteristics than collision accidents for safety analysis. For example, the occurrence of accidents is predicted by building a traffic conflict index model, such as time-to-collision (TTC), post-encroachment time (PET), and other indicators. The analysis found that the Pearson correlation coefficient between the recurrence level of millions of lane-change accidents and the number of real traffic accident was significantly correlated ($\beta = 0.77$). It proved the effectiveness of traffic conflict technology in predicting accidents [28]. Although there is no direct evidence for the predictive effectiveness of warning frequency on accidents, there is no doubt that research is necessary.

In terms of model methods, statistical models are mainly constructed based on discrete variables. Most of these studies classify the response variables reflecting collisions as categorical variables. For example, truck drivers are divided into three categories, namely, "middle-aged and elderly drivers with low risk of driving violations and high historical accident records", "high risk of driving violations and high historical accident records", and "middle-aged drivers with no record of driving violations and historical accident records" to construct an ordered logit model [29] or classify by accident severity, such as dividing truck drivers into three categories of injury levels [30]. Various logistic regression models [31,32] are commonly used models. Further studies have shown that complex models have better performance. For example, the mixed logit model has better performance than the polynomial logit model [12]. There is also a model combined with machine learning models, such as a complementary model of random parameter binary logit (RPBL) and support vector machine (SVM) [33]. The same variable has different importance in different models, and complementary models make the model more applicable. Chen et al. [30] regarded that a disadvantage of finite mixture models is that they neglect the observation heterogeneity within each data group due to the assumption of observation homogeneity in each group. A hierarchical Bayesian multinominal logit model with a random intercept setting was utilized to effectively examine attribute influence as well as the unobserved variance in multi-level data analyses. The results of the regression model can directly reflect the influence of different factors on the collision and have strong interpretability.

Compared with the frequency of traffic accidents, the risk warning of heavy construction trucks occurs more frequently and is easier to observe. However, there are no fixed routes for heavy construction trucks in the city, and the corresponding road attributes are also difficult to obtain. To this end, this paper conducted factor correlation analysis based on the warning records and trip records of a third-party supervision platform in the second half of 2021. The warning records were labelled according to the number of running trips. A regression model was established to quantitatively explain the relationship between the occurrence frequencies of various types of warnings. When there is a causal relationship between different warning factors, the early warning platform needs to consider adjusting the triggering principle of the warning factors. At the same time, it provides a reference for the formulation of the driver evaluation system.

## 3. Materials and Methods

### 3.1. Data Preprocessing

This study selected the warning records and trip records of 17 heavy construction trucks from July to December 2021. The 17 heavy construction trucks came from two small earth and stone transportation enterprises in Guangzhou and covered the intelligent supervision system according to the management measures. Through the third-party supervision platform, not only can real-time data be obtained, but historical monitoring data can also be traced back. From the platform's Evidence Center, the information in Table 1 was obtained. The table of warning records contains 20 fields, and the table of trip records contains 12 fields. Among them, there are 24 types of warning, with a total of 279,798 warning records. The details are shown in Table 2. The personal information of the

drivers of each vehicle is not available and is not considered in this paper, although they may lead to differences in behavior, such as the driver's age, gender, and driving experience.

**Table 1.** Field attribute of basic data.

| Range | Warning Records | Format | Trip Records | Format |
|---|---|---|---|---|
| 1 | Association | varchar | Association | varchar |
| 2 | License plate number | varchar | License plate number | varchar |
| 3 | License plate color | varchar | License plate color | varchar |
| 4 | SIM number | varchar | Date | date |
| 5 | Warning type | varchar | Departure time | datetime |
| 6 | Warning level | varchar | Closing time | datetime |
| 7 | Warning time | datetime | Average speed | float |
| 8 | Speed | float | Maximum Speed | float |
| 9 | Longitude | double | Is there a warning | varchar |
| 10 | Latitude | double | Path Area | varchar |
| 11 | Mileage | float | Start and End Places | varchar |
| 12 | State | varchar | Total mileage | float |
| 13 | Altitude | float | | |
| 14 | Terminal type | varchar | | |
| 15 | Car type | varchar | | |
| 16 | False warning or not | varchar | | |
| 17 | Remarks | varchar | | |
| 18 | Risk level | varchar | | |
| 19 | Risk duration | time | | |
| 20 | Risk value | float | | |

**Table 2.** Summary of warning types.

| Range | Type | Count [1] | Count without False Warning [2] | Keep or Not | Classification |
|---|---|---|---|---|---|
| 1 | Forward collision | 6944 [3] | 6944 | Yes | |
| 2 | Lane departure | 2268 | 2268 | Yes | |
| 3 | Too close distance | 19,229 | 19,229 | Yes | |
| 4 | Pedestrian collision | 3127 | 3127 | No | |
| 5 | Frequent lane changes | 0 | 0 | No | Trajectory abnormality |
| 6 | Road sign overrun | 0 | 0 | No | |
| 7 | Obstacle | 0 | 0 | No | |
| 8 | Assisted-driving fails | 0 | 0 | No | |
| 9 | Lane change across solid line | 41,788 | 41,788 | Yes | |
| 10 | Pedestrian detection in carriageway | 0 | 0 | No | |
| 11 | Driver mismatch (platform) | 0 | 0 | No | |
| 12 | Fatigue driving | 769 | 718 | Yes | |
| 13 | Answering calls | 2023 | 1982 | Yes | |
| 14 | Smoking | 5681 | 5616 | Yes | |
| 15 | Not looking ahead | 4047 | 3843 | Yes | |
| 16 | Driver abnormal | 23,438 | 23,438 | No | |
| 17 | Probe occlusion | 737 | 737 | Yes | Driving behavior abnormality |
| 18 | Driver behavior monitoring function fails | 1873 | 1873 | No | |
| 19 | Overtime driving | 0 | 0 | No | |
| 20 | Not wearing seat belt | 337 | 337 | Yes | |
| 21 | Infrared-blocking sunglasses fail | 1 | 1 | No | |
| 22 | Hands-off driving | 3400 | 3387 | Yes | |
| 23 | Playing phone | 1992 | 1992 | Yes | |
| 24 | Right rear approach | 162,144 | 162,144 | No | |

[1] Count of raw records; [2] count after deleting records of false warning identified by the system; [3] the counting units in the table are times in all trips.

In principle, the system will record the corresponding evidence as pictures or videos. After deleting the false warning records, 279,424 pieces of data remained. According to

the evidence, it was found that there were several warning factors that had no actual significance to warn, and four invalid warning factors were deleted. Specifically, most of the data of "pedestrian collision" and "driver behavior monitoring function fails" were missing. "Driver abnormal" means that the driver leaves the monitoring view due to device movement, which is meaningless. "Right rear approach" had a very high error rate in identifying objects and could only be judged manually. Eight factors without data were deleted, namely, "frequent lane changes", "road sign overrun", "obstacle", "assisted-driving fails", "pedestrian detection in carriageway", "driver mismatch (platform)", "overtime driving", and "infrared-blocking sunglasses fail". In the end, there were 12 effective warning factors, with a total of 88,841 warning records, which are divided into two categories: trajectory abnormality and driving behavior abnormality.

According to the latitude and longitude of the warning records, it was found that 22 records were not in line with reality, so they were deleted. As shown in Figure 2, QGIS3.20.2 was used to draw a spatial distribution map and heat map based on the latitude and longitude. Due to the lack of information on road attributes, spatial correlation analysis was not carried out in this study. To perform correlation analysis between different factors, it is necessary to merge the data in two tables. In this paper, a trip is taken as the sample collection unit. First, trips with no warning, trips lack of speed information, and trips with abnormal speed were deleted. The number of remaining samples was 1033. According to research experience, the number of samples should be greater than 10 times the number of variables [34]. According to the central limit theorem, each group of samples should be greater than 30. There were 1033 valid samples for 17 vehicles, and each vehicle had more than 30 samples. This sample size can be considered reasonable. The second step was to add the corresponding trip label to each warning record by comparing the license plate and time. The third step was to count the occurrences of different warning factors for each trip. In the end, three variables representing vehicle operation and 12 variables representing warning factors were obtained, and all were considered continuous variables. As mentioned in the literature review, many factors (such as traffic flow density, road type, driving speed, etc.) may have an influence on certain behaviors of drivers. During the whole trip of a heavy truck, the traffic flow density and road type are always changing. The average speed and maximum speed of the operation are considered, but other factors will not be considered in this article.

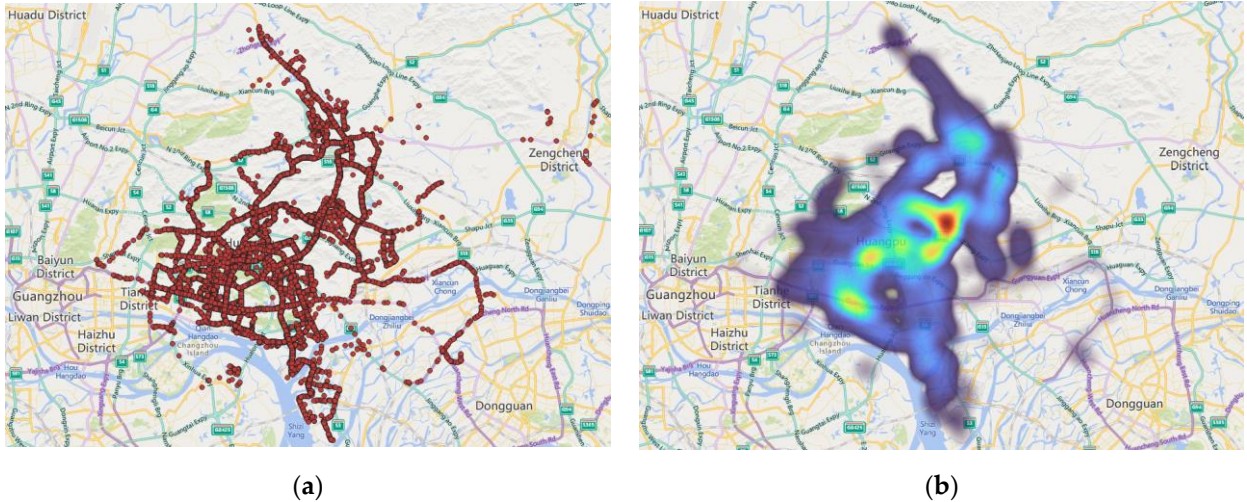

(**a**) (**b**)

**Figure 2.** (**a**) Display of alarm trigger points on the road network; (**b**) the density of alarm trigger points in space (red is the densest).

Descriptive statistical analysis was performed on the sorted data, as shown in Table 3. The median and minimum values of most warning factors were basically the same, and both were 0. From the counts of each variable, "lane change across solid line" was the

most, followed by "too close distance" and "not wearing seat belt" the least. It reflected the actual situation that the driver competed for the right of way on the premise of shortening the journey. Obviously, the warning variable contained many zero values and did not conform to the normal distribution. Therefore, the Spearman correlation coefficient was used for binary correlation analysis. The Spearman correlation coefficient uses the rank of the original value to calculate the product-difference correlation coefficient. The correlation test of these explanatory variables was carried out by SPSS 26, and the results are shown in Table 4. The results showed that the absolute values of the Spearman correlation coefficients between t1 (forward collision) and t3 (too close distance), and t3 and t4 (lane change across solid line) were greater than 0.6. That is, there was a large linear correlation between these variables. When constructing regression models, it is important to focus on collinearity between these variables to avoid significant correlations that negatively affect coefficient estimates.

**Table 3.** Descriptive statistics for the variables.

| Variables | Rename | Count [1] | Mean | SD [2] | CV [3] | Max | Min | Median |
|---|---|---|---|---|---|---|---|---|
| Tripkm [4] | Trip | - | 135.610 | 6890.924 | 2.5828 | 440.5 | 1.6 | 120.700 |
| Average speed [5] | Aves | - | 19.030 | 82.994 | 0.2834 | 42.2 | 1.3 | 17.300 |
| Max speed [5] | Maxs | - | 75.872 | 108.648 | 0.3243 | 106.3 | 34.4 | 76.900 |
| Forward collision | t1 | 4009 | 3.88 | 94.902 | 0.303 | 82 | 0 | 0.00 |
| Lane departure | t2 | 1051 | 1.02 | 16.327 | 0.126 | 52 | 0 | 0.00 |
| Too close distance | t3 | 9791 | 9.48 | 320.816 | 0.557 | 123 | 0 | 2.00 |
| Lane change across solid line | t4 | 18,345 | 17.76 | 1519.865 | 1.213 | 262 | 0 | 0.00 |
| Fatigue driving | b1 | 513 | 0.50 | 4.930 | 0.069 | 30 | 0 | 0.00 |
| Answering calls | b2 | 946 | 0.92 | 5.916 | 0.076 | 29 | 0 | 0.00 |
| Smoking | b3 | 3376 | 3.27 | 71.262 | 0.263 | 71 | 0 | 0.00 |
| Not looking ahead | b4 | 1734 | 1.68 | 42.406 | 0.203 | 54 | 0 | 0.00 |
| Probe occlusion | b5 | 656 | 0.64 | 2.620 | 0.050 | 12 | 0 | 0.00 |
| Not wearing seat belt | b6 | 127 | 0.12 | 0.271 | 0.016 | 5 | 0 | 0.00 |
| Hands-off driving | b7 | 775 | 0.75 | 49.864 | 0.220 | 144 | 0 | 0.00 |
| Playing with phone | b8 | 544 | 0.53 | 1.259 | 0.035 | 6 | 0 | 0.00 |

[1] Number of each warning behavior in each trip, the unit is times/trip; [2] standard deviation; [3] coefficients of variation, CV is the ratio of standard deviation to mean; [4] the unit is kilometers/trip; [5] the unit is kilometers/hour.

**Table 4.** Correlation matrix for the variables.

| Coef [1] | Mile | Aves | Maxs | t1 | t2 | t3 | t4 | b1 | b2 | b3 | b4 | b5 | b6 | b7 | b8 |
|---|---|---|---|---|---|---|---|---|---|---|---|---|---|---|---|
| Mile | 1 | | | | | | | | | | | | | | |
| Aves | 0.535 ** | 1 | | | | | | | | | | | | | |
| Maxs | 0.596 ** | 0.521 ** | 1 | | | | | | | | | | | | |
| t1 | 0.249 ** | 0.011 | 0.075 * | 1 | | | | | | | | | | | |
| t2 | −0.336 ** | −0.355 ** | −0.234 ** | 0.087 ** | 1 | | | | | | | | | | |
| t3 | 0.149 ** | 0.039 | 0.006 | 0.752 ** | −0.054 | 1 | | | | | | | | | |
| t4 | −0.051 | 0.022 | −0.141 ** | 0.305 ** | −0.281 ** | 0.689 ** | 1 | | | | | | | | |
| b1 | 0.154 ** | −0.037 | 0.042 | 0.299 ** | 0.260 ** | 0.185 ** | −0.195 ** | 1 | | | | | | | |
| b2 | 0.143 ** | −0.112 ** | 0.118 ** | 0.039 | −0.067 * | 0.092 ** | 0.116 ** | 0.155 ** | 1 | | | | | | |
| b3 | 0.041 | −0.209 ** | 0.019 | −0.065 * | −0.182 ** | 0.097 ** | 0.202 ** | 0.044 | 0.525 ** | 1 | | | | | |
| b4 | 0.153 ** | −0.033 | 0.032 | 0.408 ** | 0.063 * | 0.278 ** | −0.192 ** | 0.430 ** | 0.037 | 0.023 | 1 | | | | |
| b5 | 0.316 ** | 0.237 ** | 0.195 ** | 0.023 | −0.131 ** | −0.091 ** | −0.198 ** | −0.014 | −0.234 ** | −0.289 ** | 0.003 | 1.000 | | | |
| b6 | −0.016 | −0.006 | −0.065 * | −0.122 ** | −0.064 * | −0.066 * | 0.006 | −0.035 | 0.170 ** | 0.293 ** | 0.006 | −0.118 ** | 1 | | |
| b7 | 0.081 ** | 0.011 | 0.093 ** | 0.007 | −0.069 * | −0.004 | 0.050 | −0.074 * | 0.006 | −0.013 | −0.069 * | 0.019 | −0.008 | 1 | |
| b8 | 0.191 ** | 0.159 ** | 0.109 ** | −0.198 ** | −0.203 ** | −0.291 ** | −0.129 ** | −0.171 ** | −0.082 ** | −0.090 ** | −0.248 ** | 0.179 ** | −0.162 ** | 0.058 | 1 |

[1] Coefficient; ** the correlation is significant at the 0.01 level (two-tailed); * the correlation is significant at the 0.05 level (two-tailed).

### 3.2. Multiple Linear Regression

Due to the lack of research about relevant data for reference, this paper conducted a series of exploratory studies. The multiple linear regression model is used to study the relationship between dependent variables and multiple explanatory variables, and its general form is as in Formula (1). The model estimation method adopts the least square method. Multiple linear regression is based on the following assumptions:

1. There is a linear relationship between the dependent and explanatory independent variables.
2. The independent variables are not highly correlated with each other.
3. The variance of the residuals is constant.
4. The observations should be independent of one another.
5. Multivariate normality occurs when residuals are normally distributed.

$$y = \beta_0 + x_1\,\beta_1 + x_2\,\beta_2 + \ldots + x_k\,\beta_k + \varepsilon, \tag{1}$$

In Formula (1), y is the dependent or predictor variable and $x_1, x_2, \ldots, x_k$ are the explanatory variables, k is the number of explanatory variables, $\beta_1, \beta_2, \ldots, \beta_k$ are the corresponding coefficients, $\beta_0$ is the intercept, and $\varepsilon$ is the random error term.

Let $\omega = [\beta_0, \beta_1, \ldots, \beta_k]$, $X = [x_1, x_2, \ldots, x_k]$; it can also be expressed as the following matrix form:

$$y = \omega^T X + \varepsilon, \tag{2}$$

*3.3. Variable Selection*

In machine learning, variable selection, also known as feature selection, is the process of selecting a subset of relevant features from a given set of features. Among them, the attributes that are useful for the current learning task are called "relevant features", otherwise they are called "irrelevant features". The purpose of feature selection is to reduce the complexity of the model and improve the generality of the model. This paper used best subset selection and least absolute shrinkage and selection operator (Lasso) regression methods for feature selection. Feature selection mainly has the following four steps:

1. Generating a candidate set of features subsets;
2. Evaluation function to evaluate the performance of different feature subsets;
3. Setting a threshold and stopping when the evaluation function value reaches the threshold;
4. Verifying the validity of the optimal feature subset.

3.3.1. Best Subset Selection

Optimal subset selection is a basic method for independent variable selection of multiple linear regression equations. The selection process is as follows:

Step 1: Assuming that there are k features, start from the null model $M_0$ with only the intercept term.

Step 2: Fit the model with different feature combinations. When the number of variables is fixed, there are the best combinations of factors corresponding to the number of variables is achieved. Models $M_1, M_2, \ldots, M_k$ are obtained by calculating the best combination of the number of variables from 1 to k. The model is optimal when the degree of freedom adjustment complex decision coefficient ($R_a^2$) is the largest in this paper.

Step 3: Generally, the cross-check error, the Bayesian information criterion (BIC), Cp [35], or adjusted $R^2$ are used to select the optimal model among the k + 1 models obtained in step 1. In this paper, $R_a^2$ was used to determine the optimal model. The formula for $R_a^2$ is shown in Equation (3).

$$
\begin{aligned}
R_a^2 &= 1 - \frac{n-1}{n-k-1}\left(1 - R^2\right),\\
R^2 &= 1 - \frac{RSS}{TSS}\\
RSS &= \sum_{i=1}^{n}\left(y_{i-}\,\hat{y}_i\right)^2\\
TSS &= \sum_{i=1}^{n}\left(\hat{y}_i - \overline{y}\right)^2
\end{aligned}
\tag{3}
$$

In Formula (3), n is the sample size, k is the number of explanatory variables, TSS is the total sum of squares, RSS is the residual sum of squares, $y_i$ is the observations of y, $\hat{y}_i$ is the predicted value of y, and $\overline{y}$ is the mean of the observations of y.

The advantage of this method is that the principle is simple and the result is globally optimal. The disadvantage is that 2 to the power of k models need to be fitted, and the amount of calculation increases exponentially.

### 3.3.2. Lasso Regression

The Lasso regression method is a compressed estimation method proposed by [36] to replace the least squares method. Lasso regression prevents overfitting by adding an L1 penalty term to the regression coefficients, reducing the degree of variation and improving the accuracy of linear regression models. Due to the penalty value, some parameter estimation results are 0, which achieves the purpose of feature selecting. The objective function of its minimization is shown in Formula (3).

$$\min_{\omega} \frac{1}{2n} \|X\omega - y\|_2^2 + \lambda \|\omega\|_1, \tag{4}$$

In Formula (3), n is the sample size, $\lambda$ is the shrinkage parameter, and $\lambda \in (0, \infty)$. When $\lambda$ is larger, the penalty is stronger and fewer variables will be retained in the model. As $\lambda$ decreases, more and more variables are retained in the model until significant variables appear sequentially.

### 3.4. Multicollinearity Test

From the binary correlation test, we can know that there was a significant correlation between some independent variables. If an independent variable is a linear organization of one or several other independent variables, there is multicollinearity among those independent variables. To reduce the influence of multicollinearity on the coefficient estimates, we needed to test the model for collinearity. Variance inflation factor (VIF) can characterize the degree of collinearity between independent variables, and its value can reflect whether there is multicollinearity between the investigated variables. A VIF value above 10 is usually referred to as an indication that multicollinearity exists [37]. However, the VIF values of all investigated variables in the models were less than three, indicating no multicollinearity problem. The calculation formula of VIF is shown in Formula (4).

$$\text{VIF}_j = \frac{1}{1 - R_j^2}, \tag{5}$$

In Formula (4), $R_j^2 (j = 1, 2, \ldots, k)$ is the square of the complex correlation coefficient of $x_j$ to the remaining p-1 independent variables $x_i$ ($i = 1, 2, \ldots, k; i \neq j$).

From the results in Table 4, there was a weak correlation between multiple groups of variables. To further analyze the correlation between variables, a multiple regression model was constructed according to the process in Figure 3. Firstly, 12 warning variables were set as dependent variables, and the best explanatory variable combination for each dependent variable was screened out by the optimal subset method. Secondly, the adjusted R-square was used to determine the dependent variable for further analysis. Thirdly, Lasso regression was used to screen the explanatory variables for the dependent variables selected in the previous step. Finally, the VIF coefficient was used to judge the collinearity among the explanatory variables.

$$\text{adjusted } R^2 = 1 - \frac{\left(1 - R^2\right)(n - 1)}{n - k - 1}, \tag{6}$$

In Formula (6), n is the sample size, and k is the number of explanatory variables.

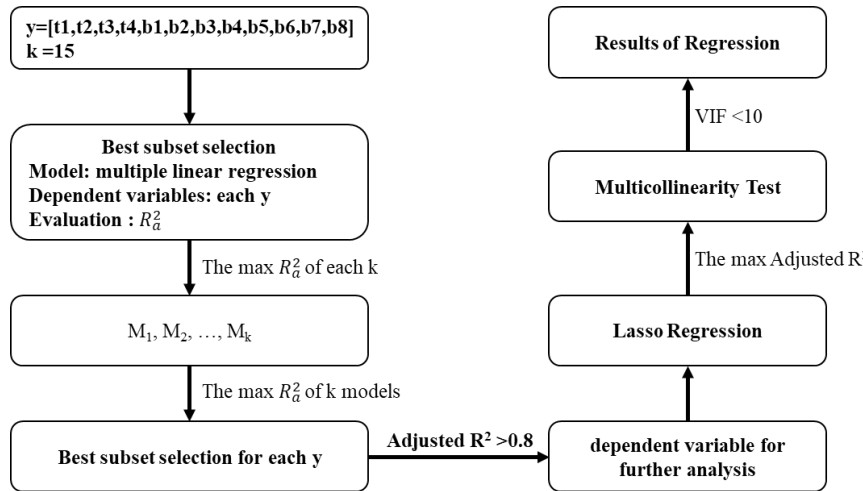

**Figure 3.** Multivariate linear equation construction process.

## 4. Results

### 4.1. Variable Selection

#### 4.1.1. Best Subset Selection

Using the optimal subset method, the multiple linear regression model was estimated and adjusted $R_a^2$ was used as the evaluation function. The estimation results of 12 warning factors as dependent variables in turn are shown in Table 5. According to the standard of an adjusted $R^2$ greater than 0.8, only t3 and t4 could construct effective multivariate linear models.

**Table 5.** Equation estimation results.

| y | Equation | Adjusted $R^2$ |
|---|---|---|
| t1 | y~Trip + Aves + t2 + t3 + t4 + b1 + b2 + b3 + b4 + b5 + b8 | 0.707 |
| t2 | y~Aves + t1 + b1 + b3 + b4 + b8 | 0.110 |
| t3 | y~Trip + Aves + Maxs + t1 + t2 + t4 + b1 + b2 + b3 + b6 | 0.895 |
| t4 | y~Trip + Aves + Maxs + t1 + t2 + t3 + b2 + b3 + b4 + b5 + b6 + b7 + b8 | 0.901 |
| b1 | y~t1 + t2 + t3 + t4 + b2 + b4 + b5 | 0.376 |
| b2 | y~Trip + Aves + Maxs + t1 + t3 + t4 + b1 + b3 + b5 + b6 + b7 | 0.291 |
| b3 | y~Trip + Aves + t1 + t2 + t3 + t4 + b2 + b6 + b7 | 0.472 |
| b4 | y~Trip + Maxs + t1 + t2 + t4 + b1 + b5 + b8 | 0.481 |
| b5 | y~Trip + t1 + t4 + b1 + b2 + b6 + b8 | 0.236 |
| b6 | y~Aves + t3 + b2 + b3 + b8 | 0.094 |
| b7 | y~t4 + b2 + b3 | 0.079 |
| b8 | y~Trip + Aves + t1 + t2 + t4 + b4 + b5 + b6 | 0.181 |

#### 4.1.2. Lasso Regression

Take t3 and t4 as dependent variables and normalize the observed values. The Lasso regression method was used to estimate the model, and the mean square error (MSE) was used as the evaluation standard. This process was implemented through the R glmnet package.

Taking t3 as the dependent variable, with the increase in λ, the degrees of freedom and residuals decreased. The coefficients of each variable decreased until they reached zero. When the variable coefficient was 0, the variable was removed. It can be seen from Figure 4a that with the increase in λ, the coefficient of the variable decreased continuously and the coefficient of some variables became 0. Figure 4b was obtained by cross-checking through the function of the glmnet package. There are two dashed lines in Figure 4b. One is the λ value when the MSE was at its minimum (λ = 0.053), and the other is the λ value when the MSE was one standard error away from the minimum MSE (λ = 0.716). The two λ values were substituted into Formula (4), and the results of the two models are shown in Table 6.

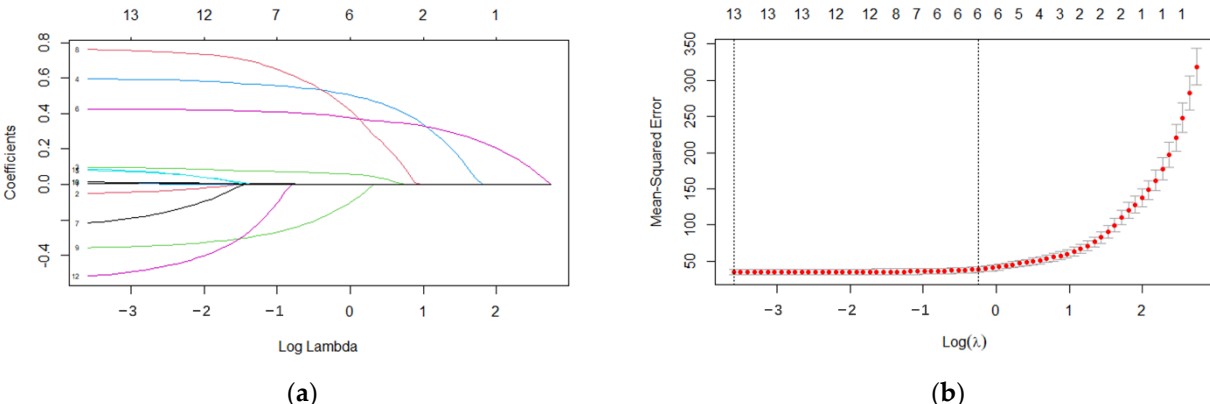

(**a**)  (**b**)

**Figure 4.** Lasso regression (y = t3); (**a**) the change of the coefficient of each variable and residual degrees of freedom of model with the increase in log(λ), 1: Trip, 2: Aves, 3: Maxs, 4: t1, 5: t2, 6: t4, 7: b1, 8: b2, 9: b3, 10: b4, 11: b5, 12: b6, 13: b7, 14: b8; (**b**) cross-check, the change of the MSE of each model and residual degrees of freedom of model with the increase of log(λ).

**Table 6.** Lasso model estimation results.

| | y = t3 | | | y = t4 | |
|---|---|---|---|---|---|
| Variables | λ = 0.053 | λ = 0.716 | Variables | λ = 0.055 | λ = 1.71 |
| Intercept | −7.031 | −4.595 | Intercept | 15.464 | 3.498 |
| Trip | 0.005 | 0.002 | Trip | −0.002 | - |
| Aves | −0.043 | - | Aves | 0.236 | - |
| Maxs | 0.095 | 0.065 | Maxs | −0.223 | −0.027 |
| t1 | 0.594 | 0.529 | t1 | −1.100 | −0.954 |
| t2 | 0.071 | - | t2 | −0.242 | - |
| t4 | 0.426 | 0.393 | t3 | 1.907 | 1.779 |
| b1 | −0.188 | - | b1 | 0.149 | - |
| b2 | 0.756 | 0.520 | b2 | −0.939 | - |
| b3 | −0.350 | −0.177 | b3 | 1.094 | 0.966 |
| b4 | 0.008 | - | b4 | −0.135 | - |
| b5 | 0.076 | - | b5 | −0.772 | −0.006 |
| b6 | −0.491 | - | b6 | −0.861 | - |
| b7 | 0.012 | - | b7 | 0.111 | - |
| b8 | - | - | b8 | −1.089 | - |
| Adjusted $R^2$ | 0.895 | 0.894 | Adjusted $R^2$ | 0.901 | 0.894 |

As shown in Figure 5a, taking t4 as a dependent variable, the smallest λ was 0.055. Similarly, the λ value when the MSE was minimum was 0.055, and the λ value of the MSE at one standard error away from the minimum MSE was 1.71, as shown in Figure 5b. The two λ values were substituted into Formula (3), and the results of the two models are shown in Table 6.

The obtained four sets of independent variables were re-estimated by the multiple linear regression model, and the adjusted $R^2$ value is shown in the last row of Table 6. Obviously, when λ = 0.053, the optimal set of independent variables of t3 was obtained, and when λ = 0.055, the optimal set of independent variables of t4 was obtained. The optimal variable set of t4 had more b1 variables compared with the selection of the best subset selection method, whereas the optimal variable set of t3 had three more variables of b4, b5, and b7.

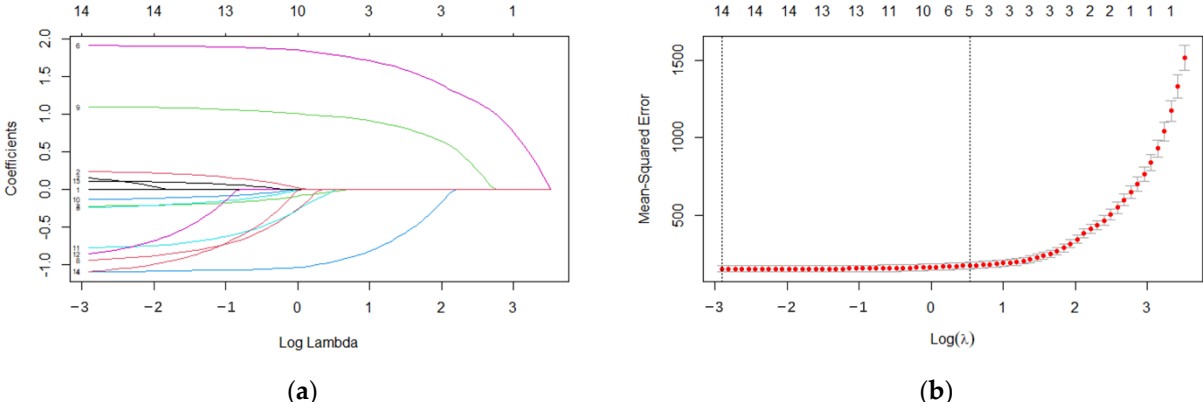

(**a**)  (**b**)

**Figure 5.** Lasso regression (y = t4); (**a**) variation of log(λ) and residual degrees of freedom, 1: Trip, 2: Aves, 3: Maxs, 4: t1, 5: t2, 6: t3, 7: b1, 8: b2, 9: b3, 10: b4, 11: b5, 12: b6, 13: b7, 14: b8; (**b**) cross-check.

### 4.2. Results of Regression

The optimal set of variables for t3 and t4 was obtained through variable selection, and the multiple linear regression was re-estimated. The results are shown in Tables 7 and 8. Obviously, although the optimal subsets obtained by the two methods were different, the final variables with significant *p*-values were the common variable of both. The *p*-values of the additional optimal variables selected by Lasso regression were not significant, including the three variables of b4, b5, and b7 in the t3 equation, and the b1 variable in the t4 equation. After removing insignificant variables, the t3 equation was obtained as Equation (7), and the t4 equation was obtained as Equation (8). The actual vs. predicted plot of Equations (7) and (8) are shown in Figure 6. It can be seen that the error of the model prediction was larger when the y value was larger.

**Table 7.** Estimation results of the multivariate linear regression model for t3.

| | | y = t3 | | | |
|---|---|---|---|---|---|
| **Variable** | **Estimate** | **S.E.** [1] | **\|t\|** | ***p* Value** [2] | **VIF** |
| Intercept | −7.403 | 1.509 | 4.906 | <0.0001 | |
| Trip | 0.006 | 0.003 | 1.853 | 0.0641 | 2.202 |
| Aves | −0.058 | 0.026 | 2.254 | 0.0244 | 1.678 |
| Maxs | 0.102 | 0.023 | 4.407 | <0.0001 | 1.775 |
| t1 | 0.597 | 0.029 | 20.800 | <0.0001 | 2.397 |
| t2 | 0.090 | 0.047 | 1.903 | 0.0573 | 1.118 |
| t4 | 0.429 | 0.006 | 69.800 | <0.0001 | 1.758 |
| b1 | −0.250 | 0.103 | 2.420 | 0.0157 | 1.615 |
| b2 | 0.770 | 0.085 | 9.010 | <0.0001 | 1.324 |
| b3 | −0.365 | 0.027 | 13.350 | <0.0001 | 1.633 |
| b4 | 0.022 | 0.039 | 0.570 | 0.5691 | 1.94 |
| b5 | 0.100 | 0.128 | 0.783 | 0.4336 | 1.308 |
| b6 | −0.547 | 0.362 | 1.512 | 0.1308 | 1.088 |
| b7 | 0.015 | 0.027 | 0.545 | 0.586 | 1.1 |
| Adjusted $R^2$ | 0.8948 | | | | |

[1] Standard error coefficient; [2] confidence interval is 95%.

The final models were tested using the 10-fold cross-validation method, and the results are shown in Table 9. MSE, root mean squared error (RMSE), and mean absolute error (MAE) were used to quantify the accuracy of prediction [38]. As shown in Table 9, the cross-validation results showed that the final regression models had certain stability for this dataset.

$$t3 = -8.15 - 0.047 \times \text{Aves} + 0.120 \times \text{Maxs} + 0.622 \times t1 + 0.429 \times t4 - 0.212 \times b1 + 0.773 \times b2 - 0.373 \times b3,$$

(7)

$$t4 = 15.518 + 0.233 \times Aves - 0.228 \times Maxs - 1.155 \times t1 - 0.224 \times t2 + 1.918 \times t3 - 0.018 \times b2 + 1.089 \times b3 - 0.826 \times b5 + 0.121 \times b7 - 1.094 \times b8, \tag{8}$$

**Table 8.** Estimation results of the multivariate linear regression model for t4.

| | | | y = t4 | | |
|---|---|---|---|---|---|
| **Variable** | **Estimate** | **S.E. [1]** | **|t|** | ***p* Value [2]** | **VIF** |
| Intercept | 15.930 | 3.189 | 4.995 | <0.0001 | |
| Trip | −0.003 | 0.007 | 0.399 | 0.6897 | 2.335 |
| Aves | 0.249 | 0.054 | 4.616 | <0.0001 | 1.665 |
| Maxs | −0.231 | 0.049 | 4.736 | <0.0001 | 1.771 |
| t1 | −1.110 | 0.064 | 17.330 | <0.0001 | 2.672 |
| t2 | −0.263 | 0.100 | 2.621 | 0.0089 | 1.131 |
| t3 | 1.911 | 0.028 | 67.980 | <0.0001 | 1.743 |
| b1 | 0.225 | 0.219 | 1.027 | 0.3045 | 1.623 |
| b2 | −0.979 | 0.185 | 5.288 | <0.0001 | 1.393 |
| b3 | 1.100 | 0.052 | 21.040 | <0.0001 | 1.337 |
| b4 | −0.148 | 0.082 | 1.806 | 0.0712 | 1.945 |
| b5 | −0.790 | 0.270 | 2.921 | 0.0036 | 1.316 |
| b6 | −0.980 | 0.775 | 1.265 | 0.2063 | 1.117 |
| b7 | 0.119 | 0.057 | 2.104 | 0.0356 | 1.096 |
| b8 | −1.155 | 0.376 | 3.067 | 0.0022 | 1.226 |
| Adjusted $R^2$ | 0.9009 | | | | |

[1] Standard error coefficient; [2] confidence interval is 95%.

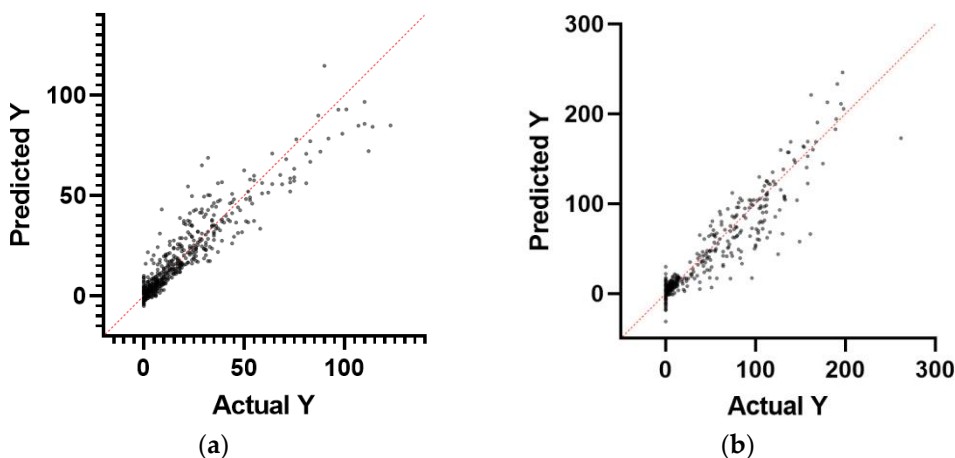

**Figure 6.** This is a figure of actual vs. predicted plots of t3 and t4. (**a**) Actual vs. predicted plot: model of t3, the result of the multiple regression model showed as Equation (7); (**b**) actual vs. predicted plot: model of t4, the result of the multiple regression model showed as Equation (8).

**Table 9.** Results of the prediction for Equations (7) and (8).

| Index | Equation (5) | | Equation (6) | |
|---|---|---|---|---|
| | **No Training Set** | **10-Fold Cross** | **No Training Set** | **10-Fold Cross** |
| MSE | 40.244 | 36.613 | 213.057 | 157.997 |
| RMSE | 6.34 | 6.051 | 14.596 | 12.570 |
| MAE | 3.20 | 3.243 | 7.345 | 7.028 |

## 5. Discussion

### 5.1. Too Close Distance

The study showed that following too closely significantly increased the odds of being involved in a crash by 1.34 times for truck drivers [26]. It is very necessary to strengthen the management of the "too close distance" behavior. According to Table 7, max speed,

t1 (forward collision), t4 (lane change across solid line), and b2 (answering calls) were positively correlated with t3 (too close distance). The largest regression coefficients were b2 (0.773), followed by t1 (0.622) and t4 (0.429). Due to multivariate effects, the positive and negative regression coefficients differed from those in the binary correlation analysis.

"Too close distance" is one of the embodiments of aggressive driving behavior [39]. Research shows that angry driving [40] and neuroticism [41] are positively correlated with aggressive driving behavior. This result showed that when drivers compete for the right of way to shorten the travel time, they will frequently illegally change lanes and accelerate frequently. The forward collision warning is determined according to the collision time between the two vehicles. When the driver ignores the forward collision warning and does not slow down, the "too close warning" will be triggered. When the driver answers the phone, the driver's attention will be distracted. The study showed that more mobile phone use causes more driving errors like speeding and collisions [42]. It is difficult to maintain a safe distance from the vehicle in front.

Average speed, b1 (fatigue), and b3 (smoking) were negatively correlated with t3. Since urban roads are often crowded, the average speed drops when vehicles travel in congested traffic. At the same time, in order not to be cut in the queue, the distance at which it is difficult to be overtaken is maintained, which often triggers an "too close distance" warning. When the driver smokes or is in a state of fatigue, the attention is relaxed, and the awareness of competing for the right of way decreases. The initiative of the vehicle in front decreases, and the situation that the distance between the vehicles is too close is reduced. If the driver of t3 often occurs in the past driving, but suddenly does not occur in one instance of driving, it is worth paying attention to whether the driver is driving while fatigued.

### 5.2. Lane Change across Solid Line

"Lane change across solid line" is also one of the embodiments of aggressive driving behavior [39]. According to Table 8, average speed, t3 (too close distance), b3 (smoking), and b7 (hands-off driving) were positively correlated with t4 (lane change across solid line). The largest regression coefficients were t3 (1.918), followed by b3 (1.089) and average speed (0.233). Solid lines are usually set on road sections where interweaving traffic should be avoided for safety, such as ramps, tunnels, entry lanes, and single lanes. These sections also often have speed limits. The study showed that higher average speeds increased risk of inattention [43]. To shorten the travel time, the driver must take the risk of being too close and having a higher speed to complete the lane change and overtaking. Surprisingly, behaviors such as "smoking" and "hands-off driving" led to more "lane change across solid line". Drivers who smoke or take their hands away while driving are always confident in their driving.

Max speed, t1 (forward collision), t2 (lane departure), b2 (answering calls), b5 (probe occlusion), and b8 (playing phone) were negatively correlated with t4. T1 ($-1.155$) had the largest negative influence on t4, followed by b8 ($-1.094$) and b5 (0.826). "Lane change across solid line" is a behavior of drivers to actively change the original trajectory, which requires a certain amount of attention to complete such planned behavior. The study showed that most non-distracted single drivers were more likely to perceive their behavior as aggressive, as opposed to distracted drivers, who were overall less likely to perceive that they drove aggressively. Obviously, behaviors such as playing phones, answering calls, and blocking probes affect the driver's concentration, and it is difficult to consider the lane changing behavior at the same time. The negative effects of "forward collision" warning and "lane departure" on t4 are difficult to explain. However, when the driver is unwilling to reduce the speed to safely follow the vehicle in front after the forward collision warning occurs, changing lanes is one way to stop the forward collision warning from continuing.

In addition, in practical work, we focus not only on the frequency of warnings, but also on whether warning events will occur. In this paper, the regression model was used to estimate the occurrence frequency of warning events. The frequency of most warnings was

0. In further research, this problem can be transformed into a binary classification problem, that is, of whether the warning event will occur.

## 6. Conclusions

Since the popularization of electrified supervision system has just begun, there is still little research on the internal correlation of warning behaviors recorded by supervision systems. The research in this paper attempts to fill the gaps. Firstly, the visualization of the spatial distribution of early warning points was carried out. Secondly, the construction of multivariate linear model and variable screening was carried out. Lane changing across a solid line is a driving violation, and if the distance between vehicles is too close, the probability of accidents will be greatly increased. The occurrence of "answering calls" and "forward collision" will be accompanied mainly by more occurrence of "too close distance". The occurrence of "too close distance" is reduced when behaviors indicative of decreased attention, such as "fatigue" and "smoking", are engaged in The occurrence of "too close distance" and "smoking" will be accompanied mainly by more occurrence of "lane change across solid line", and the occurrence of "lane change across solid line" is reduced when behaviors indicative of decreased attention, such as "probe occlusion", "answering calls", and "playing with phone", occur. Overall, distracted behaviors had a significant impact on aggressive driving behavior. Subsequent studies investigating aggressive driving behavior need to consider more historical monitoring or self-reported distracted behavior.

Based on the above results and analysis, the mechanism for triggering t3 (too close distance) and t4 (lane change across solid line) warnings should consider other relevant warning factors. For example, based on the cumulative warning frequency of related warning factors, when the predicted value of the dependent variable is greater than a threshold (such as 1), the driver can be reminded to reduce the occurrence of warnings. When formulating the driver evaluation system, the weight of the factors that bring concurrent warning should be increased. The specific implementation needs more analysis and testing. The causes of warning factors should include the driver's own characteristics and road attributes. The correlation of early warning behavior studied in this paper has information limitations.

**Author Contributions:** Conceptualization, W.Q. and S.Z.; methodology, S.Z.; software, S.Z.; validation, W.Q., S.Z. and J.H.; formal analysis, J.H.; investigation, S.Z.; resources, W.Q.; data curation, S.Z.; writing—original draft preparation, S.Z.; writing—review and editing, W.Q.; visualization, S.Z.; supervision, W.Q.; project administration, J.H. All authors have read and agreed to the published version of the manuscript.

**Funding:** This work was supported by the National Natural Science Foundation of China (No. 52072131), the Key Research Projects of Universities in Guangdong Province (No. 2019KZDXM009), the Natural Science Foundation of Guangdong Province (No. 2022A1515010123), and the Special Innovative Projects of Universities in Guangdong Province (No. 2019GKTSCX036).

**Institutional Review Board Statement:** Not applicable.

**Informed Consent Statement:** Not applicable.

**Data Availability Statement:** Not applicable.

**Acknowledgments:** The material support of company Guangdong Autotoll Intelligent Information Development Co., Ltd., Guangzhou, China, is acknowledged.

**Conflicts of Interest:** The authors declare no conflict of interest.

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
