# Peer review of "Correlation Analysis of Real-Time Warning Factors for Construction Heavy Trucks Based on Electrified Supervision System"

_sustainability, doi:10.3390/su141710944_

Round 1

Reviewer 1 Report

1. The expression "lasso regression" should be unified, rather than simply referred to as "lasso" in several places. For example, subsection titles for 2.3.2 and 3.1.2. At the same time, the text format should be checked. The “2” in line 277 should be superscript.

2. The description about Equation 4 is wrong and needs to be corrected. I.e., In the formula 4, R_j is the negative correlation coefficient between the jth variable x_j and the other variables x_i  (j=1,2,,k; ji), that is, the arithmetic square root of R2.

3.To reflect the stability of the final model, the cross test results of the final model should be supplemented.

Author Response

Concerns1:The expression "lasso regression" should be unified, rather than simply referred to as "lasso" in several places. For example, subsection titles for 2.3.2 and 3.1.2. At the same time, the text format should be checked. The “2” in line 277 should be superscript.

Response to concerns1: Relevant expressions in the full text have been unified as "lasso regression".

2.3.2 Lasso Regression

The Lasso regression method is a compressed estimation method proposed by [36] to replace the least squares method. Lasso regression prevents overfitting by adding an L1 penalty term to the regression coefficients, reducing the degree of variation and improving the accuracy of linear regression models.

Concerns2:The description about Equation 4 is wrong and needs to be corrected. I.e., In the formula 4, R_j is the negative correlation coefficient between the jth variable x_j and the other variables x_i  (j=1,2,,k; ji), that is, the arithmetic square root of R2.

Response to concerns2: The description of the formula has been modified according to the comments.

In the formula 4, is the square of the complex correlation coefficient of  to the remaining p-1 independent variables .

Concerns3:To reflect the stability of the final model, the cross test results of the final model should be supplemented.

Response to concerns3: Cross-check analysis has been added as suggested.

Table 9 Results of the prediction for equation 5 and equation 6

Reviewer 2 Report

This is an interesting study, but there are some problems in the research methods, research background, comparative analysis, and even the writing. I hope these comments will help authors improve their manuscripts.

1.     Does the sample of 17 heavy trucks meet the research needs? At the same time, does the author consider the differences of drivers (such as age, gender, driving experience, etc.)? These should be explained in the research methods.

2.     This study is related to many factors, such as traffic flow density, road type, driving speed, time, etc.. But there is no introduction or analysis of the above factors.

3.     Conclusions such as too close distance and lane change across solid line are originally some dangerous driving behaviors, and the comparison and analysis with the results of other studies are lacking in this manuscript.

4.     The language and standardization of the manuscript need to be improved, such as the naming in Figure 1, missing units in the tables, etc.

Author Response

Concerns1:Does the sample of 17 heavy trucks meet the research needs? At the same time, does the author consider the differences of drivers (such as age, gender, driving experience, etc.)? These should be explained in the research methods.

Response to concerns1: We are gratitude to you for your works and helpful comments on our manuscript. This study selected the warning records and trip records of 17 construction heavy trucks from July to December in 2021. The sample of 17 heavy trucks caused 279,798 warning records for 6 months. We assume this is a large enough sample to study behavioral patterns. Unfortunately, because the research partner is a third-party supervision platform, it does not have the right to collect the driver's personal information.

Concerns2:This study is related to many factors, such as traffic flow density, road type, driving speed, time, etc.. But there is no introduction or analysis of the above factors.

Response to concerns2: Similarly, the starting point of this paper is the optimization of the third-party supervision platform. Unfortunately, the platform only has the driving behavior alarm records and travel track records. Since heavy trucks do not have a fixed driving path, variables such as traffic flow density and road type are difficult to obtain. In this paper, operating speed, and trip distance (tripkm) were included as variables.

Concerns3:Conclusions such as too close distance and lane change across solid line are originally some dangerous driving behaviors, and the comparison and analysis with the results of other studies are lacking in this manuscript.

Response to concerns3: Too close distance and lane change across solid line are not only dangerous driving behaviors but also aggressive driving behaviors. Related comparisons and analyses have been supplemented as requested.

Concerns4:The language and standardization of the manuscript need to be improved, such as the naming in Figure 1, missing units in the tables, etc.

Response to concerns4: Sorry for not being careful enough in writing the manuscript, it has been revised as suggested.

Reviewer 3 Report

I attached my comments.

Author Response

We are gratitude to you for your works and helpful comments on our manuscript.

Reviewer 4 Report

1There should be a review on what has been studied before on this topic. Have a similar method and study performed before by other researchers? What are your new contributions? These questions are missing in the Introduction or Discussion.

2Some of the data in Table 3 suggests adding units. Also, the maximum value in Maxspeed is 199.3, is there a problem with this value.

3You need to discuss if your work is new and innovative, and what advancement you made in sustainability. There should be a review on what has been studied before on this topic. Discussions should also include some comparison with some existing works done elsewhere.

4The conclusions should be clearer.

Author Response

we are gratitude to you for your works and helpful comments on our manuscript. 

Reviewer 5 Report

To analyze the correlation between different active safety warning factors, the authors built a multiple linear regression equation associated with operational attributes and warning factors based on 17 heavy trucks’ data. This paper provided a theoretical basis for the optimization of the warning function of the electrified supervision system and the continuing education of drivers by exploring the internal correlation between different warning factors. It is a very interesting topic. Some problems list below.

1.What is the meaning of CV in Table 3?

2.Different colored line in the Figures should be clarified, or the line should be labelled in the Figures.

3.”The results showed that only t3 and t4 can construct effective multivariate linear models.”Why?

4. To build the regression model has less meaning to estimate the occurrence frequency of warning events. So the study focusing on key driving safety factors mining has more academic value.

Generally speaking, to mine the key factors affecting the construction heavy trucks’ safety is necessary and important for safe production. But this article is not innovative to publish in our journal.

Author Response

(The authors gave the same response as above.)

Round 2

Reviewer 2 Report

作者应该修改我在手稿中的评论,而不是在答案中解释它们。

1、17辆重卡的样本是否满足研究需要?同时,作者是否考虑了司机的差异(如年龄、性别、驾驶经验等)?这些都应该在研究方法中加以说明。

2. 本研究涉及交通流密度、道路类型、行驶速度、时间等诸多因素,但没有对上述因素进行介绍或分析。

在研究方法、数据分析、结果比较等方面仍需改进。

Author Response

Thank you very much for your constructive comments on our manuscript.

The manuscript has been revised according to your suggestions.

Reviewer 3 Report

The authors revised the study according to the reviewers' comments. It can be published in this version.

Author Response

We are very grateful for your decision and constructive comments on our manuscript.Your comments and suggestions are an important basis for manuscript revision.

Reviewer 5 Report

As for the authors have polished the paper carefully, and they have answered the questions correctly proposed by the reviewers, I recommend to accept it.

Author Response

Thank you for your decision and constructive comments on our manuscript.Your comments and suggestions are an important basis for manuscript revision.